# Comparison of Standard Neoadjuvant Therapy and Total Neoadjuvant Therapy in Terms of Effectiveness in Patients Diagnosed with Locally Advanced Rectal Cancer

**DOI:** 10.3390/medicina61020340

**Published:** 2025-02-14

**Authors:** Ayberk Bayramgil, Ahmet Bilici, Ali Murat Tatlı, Seda Kahraman, Yunus Emre Altintas, Fahri Akgul, Musa Barış Aykan, Jamshid Hamdard, Sema Sezgin Göksu, Mehmet Ali Nahit Şendur, Fatih Selçukbiricik, Ömer Fatih Ölmez

**Affiliations:** 1Department of Medical Oncology, Faculty of Medicine, Istanbul Medipol University, 34000 İstanbul, Türkiye; ahmetknower@yahoo.com (A.B.); jamshidhamdard@hotmail.com (J.H.); olmezof@gmail.com (Ö.F.Ö.); 2Department of Medical Oncology, Faculty of Medicine, Akdeniz University, 07000 Antalya, Türkiye; alimurat@akdeniz.edu.tr (A.M.T.); semagoksu@akdeniz.edu.tr (S.S.G.); 3Department of Medical Oncology, Faculty of Medicine, Ankara Yıldırım Beyazıt University, 06000 Ankara, Türkiye; seda.kahraman3@saglik.gov.tr (S.K.); masendur@yahoo.com.tr (M.A.N.Ş.); 4Department of Medical Oncology, Faculty of Medicine, Koc University, 34000 İstanbul, Türkiye; yunusaltintas1688@gmail.com (Y.E.A.); fselcukbiricik@kuh.ku.edu.tr (F.S.); 5Department of Medical Oncology, Faculty of Medicine, Trakya University, 22020 Edirne, Türkiye; fariak@doctor.com; 6Department of Medical Oncology, Gülhane Training and Research Hospital, University of Health Sciences, 06000 Ankara, Türkiye; musabarisaykan@gmail.com

**Keywords:** colorectal cancer, locally advanced rectal cancer, neoadjuvant therapy, total neoadjuvant therapy, standard neoadjuvant therapy, pathological complete response

## Abstract

*Background/Objectives*: The study aimed to compare the treatment effectiveness of patients with locally advanced rectal cancer undergoing standard neoadjuvant therapy or total neoadjuvant therapy. It also sought to identify prognostic factors for disease-free survival and overall survival and parameters predictive of pathological complete response. *Materials and Methods*: A retrospective analysis was conducted on 239 patients diagnosed with locally advanced rectal cancer between 2016 and 2022 at several medical centers in Turkey. Clinical data, including neoadjuvant chemoradiotherapy types, chemotherapy regimens, surgical outcomes, and survival metrics, were collected. Statistical analyses included chi-square tests, Kaplan–Meier survival analysis, and Cox proportional hazard models to evaluate prognostic factors for disease-free survival and overall survival and logistic regression to identify predictors of pathological complete response. *Results*: Among 239 patients, 46.9% received total neoadjuvant therapy, while 53.1% underwent standard neoadjuvant therapy. Total neoadjuvant therapy was associated with a significantly higher pathological complete response rate (45.5% vs. 14.9% in standard neoadjuvant therapy; *p* < 0.001) and longer disease-free survival (median 124.2 vs. 72.4 months). The 3-year overall survival rate for all patients was 90.7%, and disease-free survival was 76.8%. Multivariate analysis identified pathological complete response (HR: 2.34), total neoadjuvant therapy (HR: 5.12), and type of surgery (HR: 8.12) as independent prognostic factors for disease-free survival, and pathological complete response and absence of lymphovascular invasion as independent prognostic factors for overall survival. Logistic regression analysis showed that total neoadjuvant therapy (OR: 4.40) and initial neoadjuvant chemotherapy (OR: 2.02) were independent predictors of achieving pathological complete response. *Conclusions*: Total neoadjuvant therapy significantly improves pathological complete response rates, disease-free survival, and overall survival in patients with locally advanced rectal cancer compared to standard neoadjuvant therapy. Total neoadjuvant therapy and achieving pathological complete response are strong independent prognostic factors for both disease-free survival and overall survival, suggesting that a more intensive neoadjuvant approach may lead to better outcomes in locally advanced rectal cancer. The increased pathological complete responses rate with total neoadjuvant therapy has created an opportunity for the development of new treatment modalities and the advancement of non-surgical management strategies in the future.

## 1. Introduction

Colorectal cancer (CRC) is a common and fatal disease with increasing incidence. Worldwide, CRC is the third most frequently diagnosed cancer in men and the second cancer in women [1]. The definition of locally advanced rectal cancer (LARC) includes tumors of the T_3-4_N_x_ or T_x_N_1-2_ stage according to TNM staging. In recent years, with the effective use of neoadjuvant treatments, pathological complete response (pCR) rates have increased, overall survival (OS) and disease-free survival (DFS) durations have been prolonged, and relapse rates have decreased significantly. In these treatment modalities, the standard neoadjuvant therapy (SNT) approach was previously preferred, but in recent years the total neoadjuvant therapy (TNT) approach has entered daily practice.

Better responses were obtained in TNT, which was developed after the surgical margin positivity continued after SNT [2]. Moreover, the non-compliance of patients receiving SNT with adjuvant chemotherapy and the inability to complete the treatment have also paved the way for the adoption of TNT. With TNT, patient adherence to treatment has improved, and the risk of surgical complications delaying the continuation of therapy has been eliminated. Additionally, with the increase in pCR rates achieved through TNT, new treatment modalities that exclude surgery have also come into consideration.

In the SNT approach, the patient is first given chemoradiotherapy (CRT). In CRT, capecitabine (2 × 825 mg/m^2^) or infusional 5-FU (225 mg/m^2^) is used simultaneously with short-term (25 gy in 5 fractions) or long-term (45–50 gy in 25–28 fractions) radiotherapy (RT). After CRT is completed, the average wait is 8 weeks until surgery. After surgery, the patient is given FOLFOX (leucovorin 400 mg/m2, 5-FU 400 mg/m^2^ bolus-1200 mg/m^2^/day continuous infusion, oxaliplatin 85 mg/m^2^) or CAPEOX (oxaliplatin 130 mg/m^2^, capecitabine 1000 mg/m^2^) as an adjuvant for 12–16 weeks [3].

The risk of positive surgical margins after neoadjuvant CRT is high in patients with T4 disease, mesorectal fascia invasion, and lymph node positivity. After neoadjuvant CRT, chemotherapy (CT) administration during the waiting period until the operation provided higher resectability and pCR [2,4,5].

In TNT, neoadjuvant CRT is administered at the same dose and duration as SNT. Adjuvant CT in SNT is given after CRT in TNT. In other words, there is no adjuvant chemotherapy in TNT. Instead, FOLFOX or CAPEOX, previously administered as an adjuvant, is administered preoperatively at the same dose and for the same duration. In addition, FOLFIRINOX (leucovorin 400 mg/m^2^, 5-FU 400 mg/m^2^ bolus-1200 mg/m^2^/day continuous infusion, irinotecan 180 mg/m^2^, oxaliplatin 85 mg/m^2^) can be given as an alternative to CAPEOX and FOLFOX in patients with cT4 and mesorectal fascia invasion or threat.

In our study, 239 patients who were diagnosed with LARC, received SNT or TNT, and underwent surgery or were followed up without surgery were retrospectively evaluated. The aim of our study was to compare the treatment effectiveness of patients who underwent these treatment approaches to show prognostic factors for DFS and OS and to identify parameters that predict pCR.

## 2. Materials and Methods

### 2.1. Study Population and Variables

In total, 239 patients who diagnosed with LARC and were followed up with at the Medipol University Faculty of Medicine, the Trakya University Faculty of Medicine, the Koc University Faculty of Medicine, the Akdeniz University Faculty of Medicine, the Ankara Yıldırım Beyazit University Faculty of Medicine, and the Gulhane Training and Research Hospital between 2016 and 2022 were included in the current study. Staging of patients with an age range of 34–89 years was performed by evaluating the clinical and radiological findings at the date of diagnosis and using the AJCC/UICC TNM staging system 8th version [6].

The patients’ age at diagnosis, gender, height, weight, tumor localization in the rectum, clinical stage, neoadjuvant CRT type and duration, if applied, the type of chemotherapy received during the RT waiting period and the number of cycles, surgery type, post-surgical resection status, number of lymph nodes removed, lymphovascular invasion status (LVI), perineural invasion status (PNI), adjuvant CT type and number of cycles, metastasis status, and final status were obtained from the patients’ charts. Patients with missing clinicopathological information were excluded from the study. Patients with de novo LARC and receiving SNT or TNT were included in the study.

Patients were eligible for inclusion if they were aged 18 years or older with biopsy-proven, newly diagnosed, primary, locally advanced rectal adenocarcinoma with a distal extension of less than 16 cm from the anal verge. Local staging was performed using endorectal ultrasound or MRI. Prior to treatment, patients underwent a full colonoscopy, CT scans of the abdomen, pelvis, and chest. Patients were required to have an Eastern Cooperative Oncology Group performance status score of 0 or 1 or a corresponding Karnofsky score. Patients with a history of pelvic radiation, polyposis syndromes, inflammatory bowel disease, recurrent rectal cancer, metastatic disease, or other primary tumors within the previous 5 years were excluded. Additionally, patients with significant cardiac disease, seizure disorders, neurological conditions, psychiatric disorders, or renal, hepatic, or bone marrow dysfunction were also excluded.

In neoadjuvant CRT, capecitabine was administered to 213 patients (89.1%), infusional 5-FU was administered to 18 patients (7.5%), and long-term RT (50.4 Gy in 28 fractions) along with FOLFOX/CAPEOX was administered to 8 patients (3.4%). There were 112 patients (46.9%) in the TNT arm. In this group, 80 patients (71.4%) received CAPEOX, 31 patients (27.7%) received FOLFOX, and 1 patient (0.9%) received FOLFIRINOX.

Thirty-one patients (13%) were managed without surgery (non-operative management-NOM) because pCR was obtained by clinical and colonoscopic biopsy, while 208 (87%) patients underwent surgery.

### 2.2. Statistics

All statistical analyses were performed using SPSS version 24.0 (SPSS Inc., Chicago, IL, USA). The relationship between the presence of pCR to neoadjuvant treatment and clinicopathological factors and treatment preferences was evaluated with chi-square and Fisher’s exact tests. Survival analysis, hazard ratios (HR), and 95% confidence intervals (CI) were determined using the Kaplan–Meier method, and survival rates were compared by the log-rank test. DFS was defined as the period from the date of surgery or the date of the clinical and biopsy complete response to neoadjuvant therapy to the date of the first recurrence of the disease or the date of the last examination in cases without recurrence. OS was defined as the time from the date of diagnosis to the date of the last examination or death. Univariate analysis was performed to evaluate the importance of clinicopathological features as prognostic factors. Multivariate analysis was then performed with the Cox proportional hazards model to find independent prognostic factors for both DFS and OS. Logistic regression analysis was also applied to determine independent factors that could predict pCR. All *p* values were two-sided, and those less than or equal to 0.05 were designated as statistically significant.

## 3. Results

The 239 patients who received SNT or TNT after being diagnosed with LARC were included in the study. The median age was 60 (range: 34–89). 127 (53.1%). When we staged the patients clinically and radiologically at the time of diagnosis, 42 (17.6%) were staged as stage 2 and 197 (82.4%) were stage 3 (Table 1).

As neoadjuvant CRT, capecitabine was administered to 213 patients (89.1%). After neoadjuvant CRT, 112 patients (46.9%) received the TNT protocol, while 127 (53.1%) received the SNT protocol. Of the patients in the TNT arm, 80 (71.4%) received the CAPEOX regimen and 31 (27.7%) received the FOLFOX regimen (Table 1).

After neoadjuvant treatments, 208 patients (66.5%) were operated on, while 31 patients (13%) were managed with NOM. Of the operated patients, 159 (66.5%) underwent low anterior resection (LAR) and 49 (20.5%) underwent abdominoperineal resection (APR). When pathological staging was performed after surgery, it was observed that 43 patients (20.9%) were ypT0. ypN0 was detected in 152 (73.8%) of the patients. LVI was detected in 38 patients (19.1%), and PNI was detected in 26 patients (13.1%). A total of 70 (29%) patients achieved pCR. Of these patients, 51 (72.9%) were in the TNT arm, while 19 (27.1%) were in the SNT arm (Table 1).

While progression occurred in 49 (20.5%) of 239 patients, no progression was detected in 190 (79.5%). The clinicopathological characteristics of the patients are listed in Table 1. During a median follow-up period of 28.8 months (range; 2.33–179.0 months), the 3-year DFS rate for all patients was 76.8% (Figure 1). The median DFS of the patients was 106.2 months, and the median OS was not reached. The 3-year OS rate in all patients was found to be 90.7% (Figure 2).

When the relationship between clinicopathological factors and treatment preferences and the presence of pCR was examined, a significant relationship was found between neoadjuvant CT (*p* < 0.001), TNT protocol (*p* < 0.001), and the presence of progression (*p* = 0.034) and pCR (Table 2). pCR was significantly higher in patients receiving capecitabine in neoadjuvant CRT than in patients receiving infusional 5-FU or FOLFOX/CAPEOX. Significantly more pCR was observed in patients in the TNT arm compared to those in the SNT arm, (45.5% vs. 14.9%), respectively. Patients with pCR were significantly less likely to experience relapse or progression compared to patients without pCR (11.4% vs. 24.3%, *p* = 0.034) (Table 2).

In the univariate analysis for DFS (Table 3), the presence of TNT (*p* = 0.002), surgery type (*p* = 0.002), pT stage (*p* = 0.005), pN stage (*p* = 0.032), presence of LVI. (*p* = 0.001), presence of PNI (*p* < 0.001), administration of adjuvant CT (*p* = 0.003), and presence of pCR (*p* = 0.032) were determined as significant prognostic factors. It was observed that the median DFS duration of patients who received TNT was 124.2 months, while it was 72.4 months for received SNT. The median DFS period was found to be 138.6 months in patients who underwent LAR and 45.7 months in patients who underwent APR. When the median DFS period was examined according to post-operative pathological T staging, it was revealed that those with ypT0 had a longer DFS. When we look at the median DFS, ypT_0_ was 129.6 months. In terms of pathological N staging, the median DFS of patients with ypN0 was longer. When the median DFS was evaluated, ypN_0_ was 135.4 months. The median DFS of patients without LVI and PNI on pathology was significantly longer than those of patients with LVI and PNI (89.6 months versus 38.2 months and 79.9 months versus 21.3 months, respectively, *p* < 0.001). The median DFS of those who did not receive adjuvant CT was shorter than the patients who received it (DFS times 59.9 months versus 122.4 months, respectively, *p* = 0.003). The DFS of patients who achieved pCR with neoadjuvant therapy was significantly longer compared to patients who did not achieve pCR (median DFS, 133.1 months versus 70.8 months, respectively, *p* = 0.032, Figure 3).

Similarly, in the univariate analysis for OS (Table 4), tumor localization (*p* = 0.029), LVI (*p* = 0.005), adjuvant chemotherapy administration (*p* = 0.021), and presence of progression (*p* < 0.001) were found to be significant prognostic factors. In terms of tumor location, the median OS time of patients with tumors located in the middle rectum was 164.1 months, while those with tumors located in the upper and lower rectum were 79.1 months and 81.1 months, respectively. While the median OS time of patients without LVI in post-operative pathology was 104.6 months, it was 55.9 months for patients with positive LVI (*p* = 0.005). However, adjuvant CT was also a prognostic factor for OS. In other words, the median OS of patients who received adjuvant CT was significantly better compared to patients who did not receive adjuvant CT (144.1 months versus 75.1 months, respectively, *p* = 0.021).

Multivariate analysis was applied for OS and DFS to show whether clinicopathological factors found to be prognostic indicators in univariate analyses were independent prognostic factors (Table 5). For DFS, the presence of pCR (*p* = 0.012, HR: 2.34, 95% CI: 1.39–4.12), the presence of TNT (*p* = 0.04, HR: 5.12, 95% CI: 0.78–17.4), and type of surgery (*p* = 0.02, HR: 8.12, 95% CI: 1.29–32.2) were found to be independent prognostic factors. In other words, patients who achieved pCR had 2.34 times better DFS rate. Patients who received TNT had 5.12 times better DFS rates and patients who could undergo LAR had 8.12 times better DFS rates. However, when multivariate analysis was performed for OS, the presence of pCR (*p* = 0.006, HR: 1.16, 95% CI: 0.008–0.45) and LVI (*p* = 0.003, HR: 5.49, 95% CI: 1.79–16.8) were found to be independent prognostic indicators (Figure 4). In other words, patients with pCR had 1.16 times better OS times and those without LVI had 5.49 times better OS times (Table 5).

Logistic regression analysis was performed to identify independent factors predictive of achieving pCR (Table 6). In this analysis, administering TNT (*p* < 0.001, OR: 4.40, 95% CI: 2.25–8.59) and starting with neoadjuvant CT (*p* = 0.04, OR: 2.02, 95% CI: 1.00–4.07) were found to be independent predictive factors for pCR. In other words, receiving TNT was 4.40 times more predictive of pCR, and starting with neoadjuvant CT was 2.02 times more predictive of achieving pCR (Table 6).

## 4. Discussion

Worldwide, CRC is the third most frequently diagnosed cancer in men and the second cancer in women [1]. The most important prognostic factor after CRC resection is the pathological stage at presentation. The presence of extramural deposits, LVI, PNI, differentiation degree, preoperative CEA level, MSI, and the presence of RAS/BRAF mutations are also important for prognosis [7,8,9].

The main goal in the treatment of LARC is to reduce the risk of local recurrence and metastasis by effectively treating the primary tumor and micrometastatic disease [10]. In recent years, the most adopted protocol for this purpose was SNT. The goal of this treatment is to achieve clean resection margins or reduce tumor size to allow for sphincter-sparing surgery. Although the local recurrence rate has decreased from 30% to less than 5% with this approach, distant metastases develop in approximately 30% of patients and are still the most common cause of death [10,11]. The meta-analysis that included eight studies with 2196 LARC patients showed that TNT treatment significantly improved the pCR rate and demonstrated benefits in DFS and OS compared to standard chemoradiotherapy. TNT also significantly reduced the risk of distant metastasis [12]. In order to eliminate these problems and eradicate micrometastases more effectively, a more effective total neoadjuvant treatment approach has been developed.

Our study aimed to determine the importance of clinicopathological features in terms of prognosis in patients diagnosed with LARC who administered TNT or SNT. In addition, the goals were to find independent prognostic factors that have an impact on survival and to reveal the relationship between the presence of pCR, clinicopathological factors, and treatment preferences.

A total of 239 patients were included in our study. While 112 (46.9%) of the patients were in the TNT arm, 127 (53.1%) were in the SNT arm. The majority of patients undergoing treatment received oxaliplatin-based (CAPEOX/FOLFOX) regimens (Table 1). In this regard, our findings were consistent with the literature [5]. Patients who did not receive CT were from a period when the total neoadjuvant approach was not yet on the agenda.

pCR is of critical importance for LARC. This is because in patients who achieve pCR, better DFS and OS times have been demonstrated, and lower rates of local recurrence and metastasis are observed [13]. The identification of clinicopathological factors associated with pCR and the demonstration of predictive factors for pCR are valuable in clearly delineating the prognosis in LARC. In the study conducted by Eisar Al-Sukhni et al. [14], 23,747 patients were included to determine the factors that predict pCR, and the pCR rate of patients treated with neoadjuvant CRT and subsequent radical surgery was found to be 23.3%. Low tumor grade, low cT and cN stage, high radiation dose, and prolonging the time between the end of RT and surgery significantly increased the pCR rate. In our study, when considered in terms of relevant clinicopathological factors and treatment preferences, a significant relationship was found between neoadjuvant CT (*p* < 0.001), TNT administration (*p* < 0.001), and the presence of progression (*p* = 0.034) and pCR (Table 2). TNT increased the pCR rate. Furthermore, as expected, recurrence or progression was significantly less common in patients who achieved pCR (*p* = 0.034). In the logistic regression analysis conducted to predict pCR, the administration of TNT (*p* < 0.001, OR: 4.40, 95% CI: 2.25–8.59) and starting with neoadjuvant CT (*p* = 0.04, OR: 2.02, 95% CI: 1.00–4.07) were identified as independent predictive factors (Table 6).

In studies by Angelita Habr-Gama et al. [15] and Julio Garcia-Aguilar et al. [16], it was demonstrated that the administration of CT during the waiting period after CRT increases the rate of achieving pCR. In the study by Mojca Tuta et al. [17] comparing TNT and SNT, the pCR responses of the patients were 23% and 7%, respectively. Similarly, in the recent RAPIDO study conducted by Renu Bahadoer et al. [18], neoadjuvant CRT and preoperative CT after short-course RT were compared in patients with LARC. In this study with a high sample size, a 14% pCR rate was achieved in the arm receiving SNT, while a 28% pCR rate was achieved in the other arm (OR 2.37 [95% CI 1.67–3.37]; *p* < 0.0001). In the study, PRODIGE 23 led by Thierry Conroy et al. [19], neoadjuvant treatment with six cycles of mFOLFIRINOX followed by sequential CRT, surgery, and adjuvant CT (FOLFOX or capecitabine) was compared with SNT in terms of efficacy. In the arm where SNT was given, a pCR rate of 12% was achieved, while a pCR rate of 28% was observed in the other arm. In the CAO/ARO/AIO-12 study, patients who received CRT after neoadjuvant induction chemotherapy were compared with those who received consolidation chemotherapy after CRT in terms of pCR, and rates of 17% and 25% were obtained, respectively [20]. In our study, pCR was achieved at a rate of 29.2%. Among the 112 patients who received TNT, 51 (45.5%) achieved pCR, while among the 127 patients who received SNT, 19 (14.9%) achieved pCR (Table 2). In our study, consistent with other studies, the administration of preoperative CT (after CRT) and prolongation of the interval between RT and surgery increased the pCR rates [17,18,19]. This increases the contribution of our study to the literature by showing that CT after CRT before surgery increases pCR, even at a time when TNT is not yet standard (Table 2).

A new approach is to monitor patients with pCR detected by radiological and histopathological control after TNT without undergoing surgery. When the literature was examined for this approach, also known as “watch and wait”, the median cCR or pCR rate was seen to be 65% [21,22,23,24,25]. In a study conducted by Asoglu O. et al. [26] in our country, this was achieved in 39 patients (65%) out of 60 patients who received TNT, cCR or pCR and were followed up with NOM. In our study, out of the 112 patients who received TNT, 31 (13%) were found to have cCR and were followed up with NOM. Circumferential resection margin (CRM) involvement is a strong prognostic indicator for local recurrence. According to a study by Glynne-Jones et al., CRM involvement, which is associated with both local recurrence and poor survival, was significantly higher in patients undergoing APR [27]. Furthermore, there is also a relationship between CRM involvement in APR specimens and the removal of less tumor tissue [28]. In our study, out of the operated patients, 159 (66.5%) underwent LAR, while 49 (20.5%) underwent APR. When evaluated through univariate analysis, the DFS duration in patients who underwent LAR was more than three times longer compared to those who underwent APR (138.6 months versus 45.7 months, respectively, *p* = 0.002) (Table 3). Furthermore, in the multivariate analysis, patients who underwent LAR had an 8.12 times better DFS rate compared to those who underwent APR (*p* = 0.02) (Table 5). Our study was supportive of previous findings [27]. In patients who underwent LAR, the recurrence rate was lower, the DFS rate was higher, and postoperative comfort was better.

In the PRODIGE 23 study conducted by Thierry Conroy et al. [19], it was observed that the 3-year DFS rate for patients receiving SNT was 69%, while those additionally receiving neoadjuvant FOLFIRINOX had a rate of 75% (HR 0.69, *p* = 0.034). In this study, in the univariate analysis conducted for DFS, SNT, lower rectal location, cT4, and TNM stage 4 were identified as poor prognostic factors, while in the multivariate analysis, the addition of preoperative FOLFIRINOX to SNT and lower TNM staging were identified as positive prognostic factors. In the meta-analysis conducted by Fausto Petrelli et al. [5], it was found that patients who received TNT had better DFS (HR = 0.75, *p* = 0.1) and OS (HR = 0.73, *p* = 0.004) compared to those who received SNT. In our study, among all patients, with a median follow-up duration of 28.8 months, the 3-year DFS rate was determined to be 76.8%. The 3-year OS rate among all patients was found to be 90.7%. In the univariate analysis for DFS, significant prognostic factors that extended the DFS duration included the administration of TNT, type of surgery, pT stage, pN stage, presence of LVI, presence of PNI, postoperative administration of adjuvant CT (*p* = 0.003), and presence of pCR (Table 3). Patients who were in the TNT arm had a median DFS duration nearly twice as long as those who were in the SNT arm (124.2 months versus 72.4 months, respectively). As expected, it was revealed that patients with ypT0 and ypN0 in postoperative staging had longer DFS durations. The DFS durations of patients who achieved pCR with neoadjuvant treatment were significantly longer compared to those who did not achieve pCR (median DFS, 133.1 months versus 70.8 months, respectively, *p* = 0.032) (Table 3). Patients who received adjuvant chemotherapy had approximately twice the DFS duration compared to those who did not receive it (122.4 months versus 59.9 months). Thus, our findings were consistent with recent phase III total neoadjuvant studies [17,18,19].

The long-term 7-year results of the PRODIGE-23 trial were recently announced, showing that the neoadjuvant chemotherapy arm had DFS, MFS, and OS rates of 67.6%, 79.2%, and 81.9%, respectively. In contrast, the SNT arm had corresponding rates of 62.5%, 72.3%, and 76.1%. This was interpreted as evidence that TNT remains a more effective treatment approach in the long term.

Chang Hyun Kim et al. [29] have demonstrated the prognostic significance of LVI and PNI after neoadjuvant CRT for DFS. The 5-year DFS rate was found to be 19.3% for patients with LVI compared to 61.7% for those without LVI (*p* < 0.001). For patients with PNI, the 5-year DFS rate was 29.4%, whereas it was 68.6% for those without PNI (*p* < 0.001). The 5-year OS rates of patients without LVI are longer compared to those with LVI (83.2% versus 58.2%, respectively, *p* < 0.001). Similarly, when looking at the 5-year OS rates for PNI, patients without PNI have much longer OS rates compared to those with PNI (86.9% versus 64.3%, respectively, *p* < 0.001). In our study, similarly, the median DFS durations of patients without LVI and PNI were significantly longer compared to those with LVI and PNI (89.6 months versus 38.2 months and 79.9 months versus 21.3 months, respectively, *p* < 0.001) (Table 3). In post-operative pathology, the median OS duration of patients without LVI was 104.6 months, while it was 55.9 months for those with LVI (*p* = 0.005).

When evaluated according to tumor localization, the median OS of patients with tumors located in the mid-rectum was 164.1 months, while it was 79.1 months and 81.1 months for those with tumors in the upper and lower rectum, respectively.

According to patient feedback, the majority of patients experience clinically significant symptoms during chemotherapy and pelvic radiotherapy, with diarrhea being the most commonly reported complaint. In the PRODIGE-23 study, quality of life (QOL) was assessed, and although initially low in both arms, these scores improved over time. Similarly, the RAPIDO study demonstrated that bowel function and late toxicity outcomes were comparable between both groups. However, three years after surgery, 59% of patients in the TNT arm and 75% in the SNT arm experienced major low anterior resection syndrome. In conclusion, intensified treatment with TNT does not negatively impact quality of life compared to SNT. However, the favorable outcomes achieved with TNT do not necessarily result in an improved QOL.

The most significant limitations of our study include its retrospective design, relatively small sample size, and short follow-up duration. TNT is a newly developed treatment strategy, and the time elapsed is too short to observe late relapses or more mature survival data. Additionally, a common concern of TNT is acute toxicity to chemotherapy. Although our study did not investigate toxicity, no patients discontinued treatment due to toxicity. On the other hand, we believe that our study contributes to the literature by including patients who received TNT at a time when phase III total neoadjuvant study data were not yet available. We found that patients in this group who received CT predicted 4.4 times more pCR and better survival outcomes compared to those who did not receive CT. In addition, the inclusion of cases involving NOM, which is gradually becoming established in practice although not yet standardized, also increases the significance of our study. Furthermore, demonstrating that the total neoadjuvant approach predicts pCR and, consequently, DFS suggests a pioneering role for future studies.

In conclusion, our study demonstrates that the TNT approach, consisting of neoadjuvant CRT, neoadjuvant CT, followed by surgery and adjuvant CT, increases the rates of pCR and sphincter-saving procedures compared to the standard CRT and surgical approach. As a result, it indirectly improves patients’ quality of life and extends survival durations.

## 5. Conclusions

Today, neoadjuvant treatment has become a standard method for the management of LARC patients. With the development of neoadjuvant therapies, rates of pCR, DFS, OS, the chance for sphincter-saving surgery, and even the opportunity for NOM, have significantly increased. In our study, the effectiveness of TNT was compared with SNT, and prognostic factors were identified to predict outcomes and demonstrate their impact on survival. With the implementation of TNT, patient adherence to treatment has improved, leading to higher treatment completion rates. The completion of neoadjuvant chemotherapy has also prevented surgical complications from hindering treatment. Furthermore, the increase in complete response rates has brought non-surgical treatment options into consideration, thereby contributing to organ preservation.

In the future, there is a need for prospective and randomized studies involving a larger number of patients to compare two different TNT approaches: one with CT before neoadjuvant CRT followed by surgery and adjuvant CT and the other with CT after neoadjuvant CRT followed by surgery. Additionally, studies comparing patients who undergo NOM in these treatments are also necessary.

## Figures and Tables

**Figure 1 medicina-61-00340-f001:**
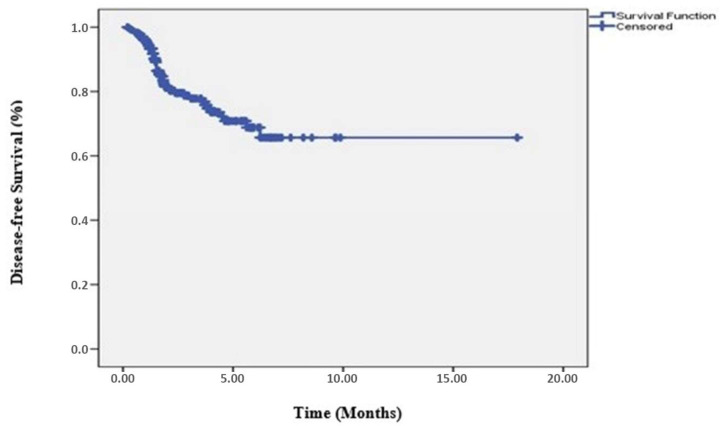
Disease-free survival in the entire patient group.

**Figure 2 medicina-61-00340-f002:**
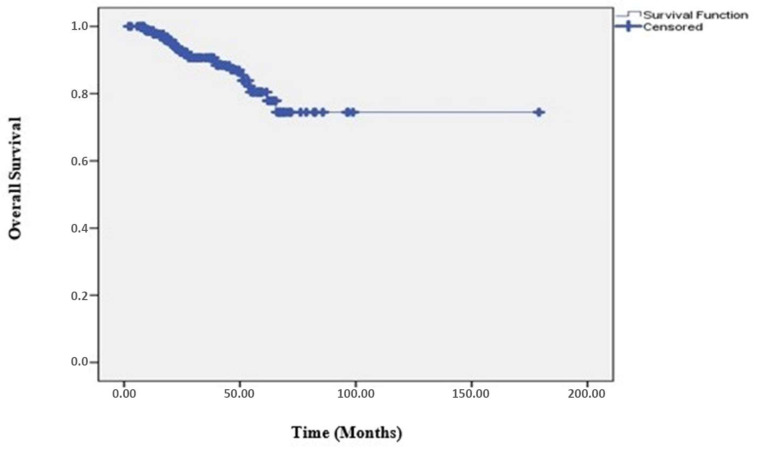
Overall survival curve.

**Figure 3 medicina-61-00340-f003:**
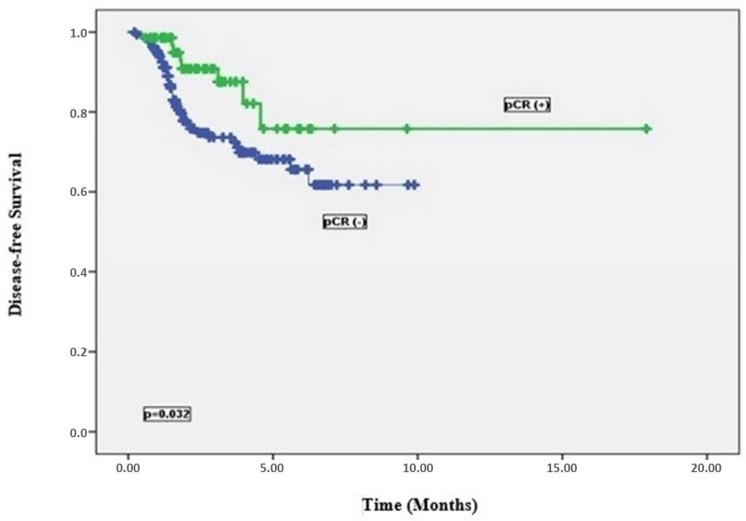
Disease-free survival curves by pCR status.

**Figure 4 medicina-61-00340-f004:**
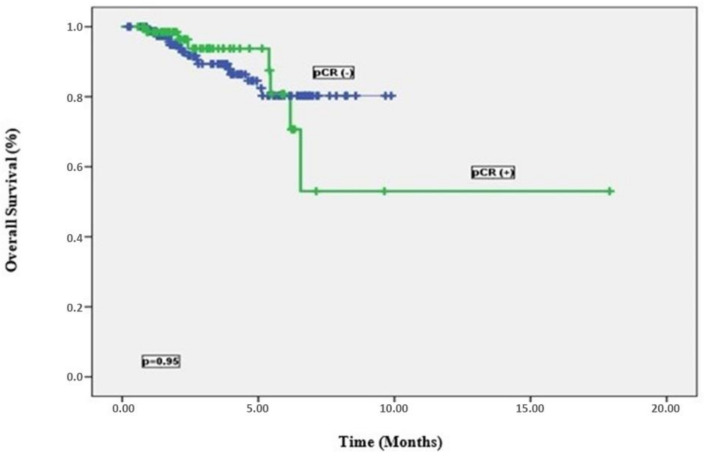
Overall survival curves by pCR status.

**Table 1 medicina-61-00340-t001:** Baseline clinicopathological characteristics of the patients.

Factors	n (%)
**Sex**	
Female	94 (39.3)
Male	145 (60.7)
**Age (Years)**	
Median, interval	60 (34–89)
≤60	127 (53.1)
>60	112 (46.9)
**Tumor localization**	
Upper rectum	22 (9.2)
Middle rectum	103 (43.1)
Lower rectum	114 (47.7)
**Clinical stage at diagnosis**	
Stage II	42 (17.6)
Stage III	197 (82.4)
**Neoadjuvant chemotherapy drug**	
Capecitabine	213 (89.1)
Infusional 5-FU	18 (7.5)
FOLFOX/CAPEOX	8 (3.4)
**TNT or SNT**	
TNT	112 (46.9)
SNT	127 (53.1)
**CT drug in the TNT arm**	
CAPEOX	80 (71.4)
FOLFOX	31 (27.7)
FOLFIRINOX	1 (0.9)
**Surgery type**	
Low Anterior Resection	159 (66.5)
Abdominopelvic Resection	49 (20.5)
Non-Operative Management	31 (13)
**Pathological T stage**	
pT0	43 (20.9)
pT1	18 (8.7)
pT2	45 (21.8)
pT3	90 (43.7)
pT4	10 (4.9)
**Pathological N stage**	
pN0	152 (73.8)
pN1	37 (18.0)
pN2	17 (8.3)
**LVI**	
No	102 (51.2)
Yes	38 (19.1)
Unknown	59 (29.7)
**PNI**	
No	115 (57.7)
Yes	26 (13.1)
Unknown	58 (29.2)
**Progression**	
No	190 (79.5)
Yes	49 (20.5)

CT: Chemotherapy, LVI: Lymphovascular invasion, PNI: Perineural invasion, SNT: Standard neoadjuvant therapy, TNT: Total neoadjuvant therapy.

**Table 2 medicina-61-00340-t002:** The relationship between the presence of pathological complete response and clinicopathological factors.

Factor	pCR n (%)	Non-pCR n (%)	*p*
**Sex**			0.47
Female	30 (57.1)	64 (37.9)	
Male	40 (42.9)	105 (62.1)	
**Age (Years)**			0.57
<60	35 (50)	92 (54.4)	
>60	35 (50)	77 (45.6)	
**Tumor localization**			0.50
Upper rectum	4 (5.8)	18 (10.7)	
Middle rectum	31 (44.9)	72 (42.6)	
Lower rectum	34 (49.3)	79 (46.7)	
**Clinical stage at diagnosis**			0.68
Stage II	14 (20)	28 (16.6)	
Stage III	56 (80)	141 (83.4)	
**Neoadjuvant chemotherapy drug**			<0.001
Capecitabine	52 (74.3)	161 (95.3)	
Infusional 5-FU	16 (22.9)	2 (1.2)	
FOLFOX/CAPEOX	2 (2.8)	6 (3.5)	
**TNT**			<0.001
Yes	51 (72.9)	61 (36.1)	
No	19 (27.1)	108 (63.9)	
**CT drug in the TNT arm**			0.094
CAPEOX	33 (62.3)	47 (79.7)	
FOLFOX	19 (35.8)	12 (20.3)	
FOLFIRINOX	1 (1.9)	0 (0)	
**Surgery type**			0.42
Low Anterior Resection	33 (47.2)	126 (74.5)	
Abdominopelvic Resection	6 (8.6)	43 (25.5)	
Non-Operative Management	31 (44.2)		
**Progression**			0.034
No	62 (88.6)	128 (75.7)	
Yes	8 (11.4)	41 (24.3)	

CT: Chemotherapy, TNT: Total neoadjuvant therapy.

**Table 3 medicina-61-00340-t003:** Prognostic factors for disease-free survival by univariate analysis.

Factors	DFS Median (Months)	*p*
**Sex**		0.49
Female	75.6	
Male	122.3	
**Age (Years)**		0.50
≤60	73.2	
>60	125.0	
**Tumor localization**		0.51
Upper rectum	70.1	
Middle rectum	140.1	
Lower rectum	69.6	
**Clinical stage at diagnosis**		0.20
Stage II	61.1	
Stage III	117.7	
**Neoadjuvant chemotherapy drug**		0.80
Capecitabine	134.2	
Infusional 5-FU	NR	
FOLFOX/CAPEOX	NR	
**TNT**		0.002
Yes	124.2	
No	72.4	
**CT drug in the TNT**		0.75
CAPEOX	NR	
FOLFOX	NR	
FOLFIRINOX	NA	
**Surgery type**		0.002
Low Anterior Resection	138.6	
Abdominopelvic Resection	45.7	
**Pathological T stage**		0.005
pT0	129.6	
pT1	78.7	
pT2	62.1	
pT3	59.2	
pT4	20.5	
**Pathological N stage**		0.032
pN0	135.4	
pN1	56.2	
pN2	44.2	
**LVI**		0.001
No	89.6	
Yes	38.2	
Unknown	59.2	
**PNI**		<0.001
No	79.9	
Yes	21.3	
Unknown	67.3	
**pCR**		0.032
No	70.8	
Yes	133.1	

CT: Chemotherapy, DFS: Disease-free survival, LVI: Lymphovascular invasion, pCR: Pathological complete response, PNI: Perineural invasion, SNT: Standard neoadjuvant therapy, TNT: Total neoadjuvant therapy, *NA: Not available; NR: Not reached*.

**Table 4 medicina-61-00340-t004:** Prognostic factors for overall survival by univariate analysis.

Factors	OS Median (Months)	*p*
**Sex**		0.29
Female	92.1	
Male	142.2	
**Age (Years)**		0.61
≤60	89.5	
>60	144.1	
**Tumor localization**		0.029
Upper rectum	79.1	
Middle rectum	164.1	
Lower rectum	81.1	
**Clinical stage at diagnosis**		0.29
Stage II	78.5	
Stage III	137.3	
**Neoadjuvant chemotherapy drug**		0.28
Capecitabine	148.1	
Infusional 5-FU	NR	
FOLFOX/CAPEOX	NR	
**TNT**		0.52
Yes	1150.6	
No	88.9	
**CT drug in the TNT**		0.50
CAPEOX	112.4	
FOLFOX	NR	
FOLFIRINOX	NA	
**Surgery type**		0.17
Low Anterior Resection	150.4	
Abdominopelvic Resection	76.1	
**Pathological T stage**		0.66
pT0	129.4	
pT1	83.7	
pT2	74.2	
pT3	87.2	
pT4	62.9	
**Pathological N stage**		0.37
pN0	153.2	
pN1	72.8	
pN2	65.1	
**LVI**		0.005
No	104.6	
Yes	49.9	
Unknown	69.8	
**PNI**		0.29
No	94.7	
Yes	55.9	
Unknown	88.1	
**pCR**		0.95
No	97.2	
Yes	148.8	

CT: Chemotherapy, LVI: Lymphovascular invasion, OS: Overall survival, pCR: Pathological complete response, PNI: Perineural invasion, SNT: Standard neoadjuvant therapy, TNT: Total neoadjuvant therapy, *NA: Not available; NR: Not reached*.

**Table 5 medicina-61-00340-t005:** Prognostic factors for disease-free survival and overall survival by multivariate analysis.

Factors					
**DFS**	**ß**	**X2**	** *p* **	**HR**	**CI 95%**
pCR	2.71	5.33	0.012	2.34	1.39–4.12
TNT	1.21	2.33	0.04	5.12	0.78–17.4
ypT stage	−0.29	0.13	0.71	0.74	0.15–3.53
ypN stage	−0.15	0.04	0.83	0.86	0.21–3.43
LVI	1.70	2.93	0.08	5.48	0.78–28.3
PNI	1.36	1.68	0.19	3.90	0.49–30.4
Surgery type	2.09	4.98	0.02	8.12	1.29–32.2
Adjuvant therapy	10.48	0.01	0.97	0.88	0.12–4.56
**OS**					
pCR	−2.81	7.41	0.006	1.16	0.008–0.45
TNT	0.63	1.57	0.21	1.89	0.69–5.12
LVI	1.70	8.86	0.003	5.49	1.79–16.8
Rectum tumor localization	0.16	0.20	0.64	1.18	0.57–2.41
Adjuvant therapy	−0.14	0.069	0.79	0.86	0.30–2.47

DFS: Disease-free survival, LVI: Lymphovascular invasion, OS: Overall survival, pCR: Pathological complete response, PNI: Perineural invasion, TNT: Total neoadjuvant therapy. CI, Confidence intervals; HR, Hazard ratio (relative risk).

**Table 6 medicina-61-00340-t006:** Regression analysis for factors predicting pathological complete response.

Factors	Beta	X^2^	*p*	OR	CI 95%
TNT	1.48	18.8	<0.001	4.40	2.25–8.59
Neoadjuvant CT	0.70	3.89	0.04	2.02	1.00–4.07
Neoadjuvant RT duration	1.19	1.34	0.24	3.30	0.43–24.9
Rectum tumor localization	0.19	0.61	0.43	1.21	0.74–1.97
Clinical stage	−0.53	2.97	0.08	0.58	0.31–1.07
Age	0.006	0.44	0.50	1.00	0.98–1.02
Sex	0.36	1.28	0.25	1.43	0.76–2.68

CT: Chemotherapy, TNT: Total neoadjuvant therapy, RT: Radiotherapy, CI, Confidence intervals; OR (odds ratio): relative risk of showing pathological response.

## Data Availability

Data available on request due to restrictions.

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
