# Peer review of "Comparison of Standard Neoadjuvant Therapy and Total Neoadjuvant Therapy in Terms of Effectiveness in Patients Diagnosed with Locally Advanced Rectal Cancer"

_medicina, 2025, doi:10.3390/medicina61020340_

Round 1

Reviewer 1 Report

Comments and Suggestions for Authors

• Do you consider the topic original or relevant to the field? Does it address a specific gap in the field? 

The modern management of rectal cancers continues to evolve. With the release of data from new landmark randomized controlled trials (RAPIDO, PRODIGE-23), total neoadjuvant therapy (TNT) has moved to the forefront of locally advanced rectal cancer treatment. It is considered today as the gold standard in locally advanced rectal cancers.

• What specific improvements should the authors consider regarding the methodology? What further controls should be considered?

A clear demonstration of both clinical and pathological complete response are mandatory. Yet, the current challenge is to accurately select patients with an apparent clinical complete response, based on clinical assessment, who would be found to have a pathological complete response if they were to undergo resection. 

• Are the conclusions consistent with the evidence and arguments, and do they address the central question? 

Yes, the conclusions are consistent with the data of the studied group. In patients with adjuvant chemotherapy, the disease-free survival is more than double compared to patients who did not receive adjuvant chemotherapy (122,4 vs. 59,9 months). This is opposite to some other studies that seem to demonstrate a better survival of patients with tumors located in the upper rectum vs those with tumors located in the mid and the inferior parts of the rectum

• The references are appropriate and recent; about half are under five years old. 

•  There is no comment about the tables and figures. 

Author Response

Response: Thank you for reviewing our study. The necessary reviews have been conducted, and changes have been implemented.

Reviewer 2 Report

Comments and Suggestions for Authors

The manuscript provides valuable insights into the comparison of total neoadjuvant therapy (TNT) and standard neoadjuvant therapy (SNT) for locally advanced rectal cancer (LARC), with a clear focus on prognostic factors and treatment outcomes. While the study is methodologically robust and well-organized, some areas could be improved for clarity and impact. The abstract should emphasize the clinical significance of findings and avoid unexplained abbreviations. The introduction would benefit from a more detailed explanation of the rationale for TNT and the challenges with SNT. Additional clarity on inclusion criteria and handling of missing data in the methods section is needed. Results are well-presented, but integrating data from tables and figures into the discussion would enhance their relevance. The discussion could be expanded to address the implications of TNT on long-term outcomes, quality of life, and its alignment with recent clinical trials, while also acknowledging study limitations. Figures and tables should include clearer annotations and be more thoroughly referenced in the text. The conclusion should reinforce the clinical significance of TNT’s benefits, and minor language refinements throughout the manuscript would improve readability. Overall, the manuscript makes a strong contribution to the field but could benefit from these refinements for greater clarity and impact.

Author Response

Comments 1: The abstract should emphasize the clinical significance of findings and avoid unexplained abbreviations.

Response 1: The abbreviations have been removed from the abstract. The clinical significance of the findings has been discussed in the abstract.

Comments 2: The introduction would benefit from a more detailed explanation of the rationale for TNT and the challenges with SNT.

Response 2: The rationale for TNT has been explained in the introduction section. The challenges of SNT have also been discussed.

Comments 3: Additional clarity on inclusion criteria and handling of missing data in the methods section is needed.

Response 3: The inclusion and exclusion criteria have been described in detail in the methods section.

Comments 4: Results are well-presented, but integrating data from tables and figures into the discussion would enhance their relevance.

Response 4: Tables and figures were more prominently included in the discussion section.

Comments 5: The discussion could be expanded to address the implications of TNT on long-term outcomes, quality of life, and its alignment with recent clinical trials, while also acknowledging study limitations.

Response 5: The long-term outcomes of TNT and quality of life were discussed. The limitations of the study were also stated.

Comments 6: Figures and tables should include clearer annotations and be more thoroughly referenced in the text.

Response 6: The explanations of the tables and figures have been expanded, and more frequent references have been made in the text.

Comments 7: The conclusion should reinforce the clinical significance of TNT’s benefits, and minor language refinements throughout the manuscript would improve readability.

Response 7: In the conclusion section, the benefits of TNT have been discussed in more detail. Minor language corrections were made.